# Expanding protected area coverage for migratory birds could improve long-term population trends

Jennifer A. Border [1] ✉, James W. Pearce-Higgins [1,2,3], Chris M. Hewson [1], Christine Howard [4], Philip A. Stephens [4], Stephen G. Willis [4], Richard A. Fuller [5], Jeffrey O. Hanson [6], Henk Sierdsema[7], Ruud P. B. Foppen[8], Lluís Brotons[9,10,11,12], Gabriel Gargallo[11,12], Daniel Fink[13] & Stephen R. Baillie [1]

Populations of many migratory taxa have been declining over recent decades. Although protected areas are a cornerstone for conservation, their role in protecting migratory species can be incomplete due to the dynamic distributions of these species. Here, we use a pan-European citizen science bird occurrence dataset (EurobirdPortal) with Spatiotemporal Exploratory Modelling to assess how the weekly distributions of 30 passerine and near passerine species overlap with protected areas in Europe and compare this to range adjusted policy protection targets. Thirteen of our 30 species were inadequately covered by protected areas for some, or all, of the European part of their annual cycle under a target based on the 2020 Convention on Biodiversity framework and none were adequately covered under a target based on the 2030 Convention on Biodiversity framework. Species associated with farmland had the lowest percentage of their weekly distribution protected. The percentage of a species' distribution within protected areas was positively correlated with its long-term population trend, even after accounting for confounding factors, suggesting a positive influence of protected areas on long-term trends. This emphasises the positive contribution that an informed expansion of the European protected area system could play for the future conservation of migratory land birds.

Migration is the persistent, directional movement of an organism from one area to another, involving specific departure and arrival behaviours[1]. As a strategy it enables individuals to respond to environmental seasonality[2,3], but can also be associated with significant costs[4]. Migratory animals make a major and unique contribution to biodiversity and ecosystem function, both as transient foragers and food sources and as transporters of energy, nutrients, propagules, toxicants, parasites and pathogens[5]. In this way, migratory animals provide predictable pulsed changes in energy flow, food web dynamics and community structure[5]. However, many migratory species have

[1]British Trust for Ornithology, The Nunnery, Thetford, UK. [2]Conservation Science Group, Department of Zoology, Cambridge University, The David Attenborough Building, Pembroke Street, CB2 3QZ Cambridge, UK. [3]School of Biological Sciences, University of East Anglia, NR4 7TU Norwich, UK. [4]Conservation Ecology Group, Department of Biosciences, Durham University, Durham, UK. [5]School of the Environment, University of Queensland, Brisbane, QLD, Australia. [6]Department of Biology, Carleton University, Ottawa, ON, Canada. [7]Sovon Dutch Centre for Field Ornithology, Nijmegen, The Netherlands. [8]Radboud Institute for Biological and Environmental Sciences (RIBES), Radboud University, Nijmegen, The Netherlands. [9]CSIC, Cerdanyola del Vallès, Spain. [10]CREAF, Cerdanyola del Vallès, Spain. [11]Catalan Ornithological Institute (ICO), Barcelona, Spain. [12]European Bird Census Council (EBCC), Prague, Czechia. [13]Cornell Lab of Ornithology, Ithaca, NY, USA. ✉e-mail: jennifer.border@bto.org

been declining over recent decades. This is thought to be due to human activities such as habitat destruction, the creation of barriers, over-exploitation and climate change[6,7], causing knock-on effects on ecosystem function and services[5].

Under the 1979 Convention on Migratory Species, countries have a duty to protect migratory animals, conserve or restore the places they inhabit and mitigate obstacles to migration[8]. Protecting areas of land and sea is considered a fundamental cornerstone for conservation[9,10] and, when well-managed, can reduce habitat loss, maintain species richness, occurrence and abundance[11,12], and offer protection from threats, such as hunting[13]. The 10[th] Conference of the Parties to the Convention on Biological Diversity in 2010, at Nagoya, aimed to protect 17% of the earth's land and inland water by 2020[14]. This target was considered achieved[15] and replaced with a new more ambitious target to protect 30% of the earth's lands, oceans, coastal areas and inland water by 2030 (referred to as 30 by 30)[16]. In Europe, the EU biodiversity strategy for 2030[17] promises to enlarge the existing Natura 2000 network of protected areas as part of progress towards 30 by 30, ensuring strict protections for areas of high biodiversity.

Due to the dynamic distributions of migratory species through the year, ensuring sufficient protection across all parts of their lifecycle is challenging. A threat in any part of a migrant's annual cycle can impact the entire population[18], particularly at times when the range is more condensed. But few migratory species have well-mapped annual distributions[18] and, therefore, assessing progress in protecting their dynamic distributions is difficult. An initial attempt to quantify this for 1451 birds using range maps of breeding, passage and non-breeding grounds found that only 9% of migratory birds are adequately protected across all stages of their annual cycle, compared to 45% of non-migratory birds[19]. However, range maps are typically of coarse resolution and do not account for fine-scale variations in space and time. Therefore, assessments of protected area coverage using these methods are likely to be inaccurate, potentially over or underestimating protection[20]. For protected areas to be effective in conserving migratory species, key habitats and resources at all stages of their life cycles need to be adequately protected[18,19].

How much of a species range needs to be protected to ensure population security and persistence (i.e. adequate protection) is more difficult to determine. Some studies have used a flat rate such as 30% to assess adequate coverage, e.g. [21]. Others have scaled coverage according to range size[19,22,23]. Specific targets are generally arbitrary and although it is biologically sensible to protect a higher proportion of the range if the range is smaller[23], the exact values do not have a biological basis. Many previous studies used a minimum requirement of 10% of a species' range protected[19,22,23], following an approach first suggested by Rodrigues et al.[23]. This threshold was selected as it approximated the proportion of the world's land surface covered by protected areas at the time, and therefore widespread species should be considered adequately protected under this threshold, as long as there are no systematic biases in the protected area network[23]. Now, however, it is widely considered that 10% protected area cover is insufficient[22,24,25]. An alternative approach to determining how much of a species range to protect uses the International Union for Conservation of Nature (IUCN) criteria, including ensuring a population will not decline by more than 30% over 10 years[25,26]. This method generally produces much larger protected area targets than other approaches (~60% of the planet protected[26]), which are unlikely to be feasible in the near term and, therefore, are not useful for guiding conservation decisions over the next decade. Setting targets for protected area coverage of species' ranges has therefore aimed to balance what is potentially achievable and what is needed to maintain sustainable populations.

Here, we use Spatiotemporal Exploratory Models (STEM)[27] and fine-scale bird occurrence data from EuroBirdPortal (https://eurobirdportal.org) to estimate the weekly European distributions of 30, passerine and near passerine, African-Palearctic migratory

landbirds for the period when they occur in Europe and assess how well they are covered by protected areas. Because challenges remain in identifying the minimum proportion of a species that should be protected, we primarily examined biases in species' representation by protected areas based on the percentage of each species' weekly distributions covered by protected areas. We also assessed shortfalls in species' representation using two target setting approaches, based on Rodrigues et al.[23], with adjustments using protected area coverage targets described in the 2020[14] and 2030[16] Convention on Biological Diversity (CBD) frameworks. To avoid setting overly stringent targets when a species is entering or leaving Europe, we only include weeks where the sum total of a species' probability of occupancy across the study area is equal to or greater than 25% of the maximum summed occurrence recorded for the species (over the whole season). By considering these two target setting approaches, we explore how well Europe has achieved protection of migratory species based on 2020 policies, and how much progress remains for meeting 2030 policies. We uncover gaps in the current protected area network with respect to migrant birds, particularly for farmland species. However, species still have more positive population trends with higher protected area cover, indicating that expanding the current network to cover some of these gaps would be beneficial. Since our work assesses migratory species' representation on a weekly basis and explicitly accounts for their mobility, our findings provide unique insight into the level of protection that migratory species experience during their migration and reveals gaps in the current protected area system.

## Results

### Species distribution models

Maps of each species' modelled distributions per week can be found at https://doi.org/10.5281/zenodo.10960419. The Area Under the Curve (AUC) of the STEM for each species ranged from excellent (0.829 for European Nightjar *Caprimulgus europaeus*) to outstanding (0.954 for Collared Flycatcher *Ficedula albicollis*) discrimination[28] (Supplementary Table 1, $n = 4,874,228$). The high performance of the models means we can confidently use them to examine spatiotemporal patterns in protected area representation.

### Protected area coverage

Overall, 20.1% of our study area was covered by protected areas. Although this coverage meets the 2020 CBD criteria[14], it falls short of the 2030 CBD criteria[16] of 30% land in a protected area. Among the locations covered by protected areas, 28.5% were covered by category I to IV protected areas, 20.5% by category V and VI protected areas, and 51.1% by protected areas that were either not assigned to an IUCN category or the category was not reported.

The percentage of species summed occurrence protected varied from a maximum of 46.6% protected area cover for Ring Ouzel *Turdus torquatus* in week 13 to a minimum of 5.3% for Ortolan Bunting *Emberiza hortulana* in week 33 (Fig. 1a). Mean protected area cover over all weeks was again highest for Ring Ouzel (34.9%) and lowest for Ortolan Bunting (9.3%). All of our species except Nightjar (in week 32), Ortolan Bunting (in weeks 31–35) and Ring Ouzel (in week 33) had weekly distributions over 250,000 km². For every week, we compared protected area coverage against targets of 17% and 30% (or range adjusted increases in these values for the weeks where a species' range was below 250,000 km²). Apart from Ortolan Bunting, all species were adequately protected in at least some weeks (mean = 17.8 ± 1.72 weeks) under the 17% target, but only Ring Ouzel (in 20 weeks), Nightjar (in one week) and Collared Flycatcher (in one week), were adequately protected in one or more weeks under the 30% target. Thirteen of the 30 species were not adequately covered by protected areas throughout their time in Europe (Fig. 1a) under the 17% target and none of the 30 species were adequately covered throughout their time in Europe under the 30% target (Fig. 1a). If excluding protected areas classified as

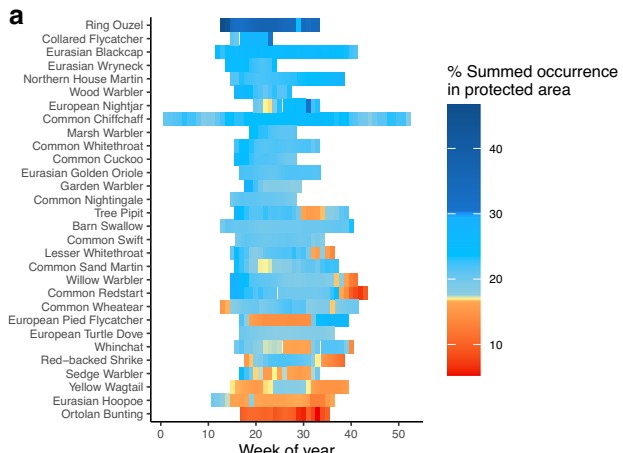

**Fig. 1 | Coverage of species by protected areas.** The percentage of each species summed occurrence in protected areas per week (**a**) including all International Union for Conservation of Nature (IUCN) categories of protected area and undesignated ($n = 633$), **b** including just IUCN categories I–IV and undesignated ($n = 633$). Species are ordered based on mean coverage of protected areas taken over all weeks the species was considered in Europe (i.e. for all weeks when the summed occurrence in Europe was ≥25% of the maximum summed occurrence). The colours denote the differences in percentage protected area cover between species relative to our 17% and 30% targets. The same colour scale is used for both graphs (i.e. 15% protected area cover is the same shade of orange). The dataset used in this figure is available in Source Data.

category V and VI which have been argued to be ineffective[29,30], only Ring Ouzel is adequately protected under the 17% target (Fig. 1b).

The percentage of a species summed occurrence covered by a protected area was significantly related to a species' weekly range size and the time of year ($n = 625$ in 30 groups by species). Species with larger ranges had a slightly higher percentage covered by protected areas (back-transformed from scaled $ß = 4.22 \times 10^{-7} \pm 1.40 \times 10^{-7}$ (SE), $p = 0.0025$). Protected area coverage was higher during spring passage (mean percentage of a species summed occurrence covered by a protected area = $22.2 \pm 0.79$ %, $p < 0.001$) compared to $20.3 \pm 0.78$% for autumn passage $p < 0.001$ and $20.7\% \pm 0.76$ for the breeding season $p < 0.001$.

Protected area coverage varied between species according to habitat association ($n = 30$, df = 27, $p < 0.001$). Farmland species had a significantly lower percentage of their summed occurrence in a protected area (mean = $17.4 \pm 1.31$% protected) compared to 'other' species (Tukey pairwise comparison: $p = 0.019$, mean = $22.1 \pm 0.93$% protected) but not to forest species (Tukey pairwise comparison: $p = 0.089$, mean = $21.8 \pm 1.51$% protected). Protected area coverage did not differ between forest and 'other' species (Tukey pairwise comparison: $p = 0.989$).

Species with a higher proportion of their summed occurrence within protected areas had more positive long-term population trends (Fig. 2, $ß = 0.052 \pm 0.015$, $n = 28$, df = 25, $p = 0.0019$, population trend as a log ratio). This significant positive effect of protected area cover remained when including either habitat ($ß = 0.044 \pm 0.017$, $n = 28$, df = 23, $p = 0.0165$), migration distance ($ß = 0.042 \pm 0.018$, $n = 28$, df = 24, $p = 0.0292$) or log body mass ($ß = 0.057 \pm 0.014$, $n = 28$, df = 24, $p < 0.001$) as additional variables. We also repeated the analysis removing the three short distance migrants (in blue on Fig. 2) and still obtained a significant positive relationship between protected area and species trend ($ß = 0.067 \pm 0.020$, $n = 25$, df = 22, $p = 0.0026$). There was no credible evidence that mean range size for each species was related to long-term trend ($ß = 0.073 \pm 0.067$, $n = 28$, df = 25, $p = 0.288$).

## Discussion

We modelled the weekly distributions of 30 species of passerine and near passerine Afro-Palearctic migrant landbirds during their occurrence in Europe. We assessed, on a weekly basis, whether these dynamic distributions met range adjusted targets based on the 2020 (17% minimum) and 2030 (30% minimum) CBD targets for protected

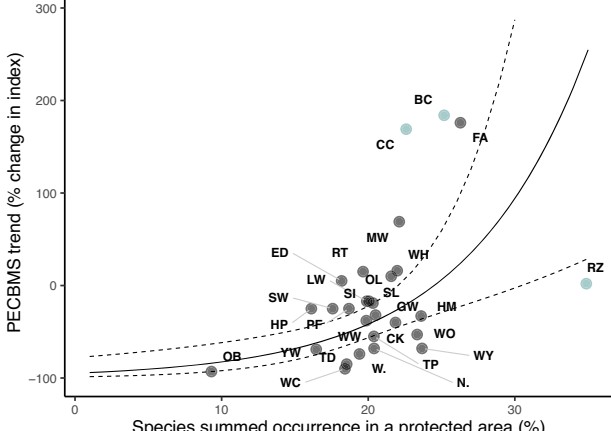

**Fig. 2 | Species population trend versus protected area cover.** Relationship between 30-year Pan-European Common Bird Monitoring Scheme (PECBMS) trend and the average percentage of a species summed occurrence in a protected area in Europe (over all weeks where the species summed occurrence is equal to or above 25% of the maximum summed occurrence). The dots are the raw data, the solid line is the prediction (back-transformed to percentage change from log ratio) from the linear regression model of trend against protected area coverage ($ß = 0.052 \pm 0.015$, $n = 28$, df = 25, $p = 0.0019$) and species range size when range size is set to the mean value across all species (3,619,119 km²). The dots are labelled with species codes which can be found in Supplementary Table 2 and coloured according to migration strategy with short-distance migrants in blue and long-distance migrants in grey. The dotted lines are the upper and lower 95% confidence intervals. The dataset used in this figure is available in Source Data.

area coverage. On a land cover basis, we found that our study area fell short of the 2030 CBD target of 30% protected area cover[16] but met the 2020 CBD target of 17% protected area cover[14]. At the species level (assuming that individuals are evenly distributed within a 10 km square with respect to the location of protected areas), currently, no species was adequately covered by protected areas under our 30% target throughout the European stage of their lifecycle, although Ring Ouzel was close to achieving this. Thirteen of 30 species were inadequately covered under our 17% target during at least some weeks of the year. If we discount protected areas known to be in categories V and VI, as

these have less stringent protection[29,30], only Ring Ouzel is adequately protected under our 17% target. Species' protected area coverage was highest during spring passage and amongst species with larger range sizes, but the effect was small. Farmland birds had a lower proportion of their summed occurrence covered by protected areas. Species' long-term trends were positively correlated with the proportion of their summed occurrence that overlapped with protected areas, suggesting a positive effect of protected areas on conserving populations which was still evident when accounting for range size, migratory distance, body mass and habitat preference. Although we focus on Afro-Palearctic migrant birds, which breed in Europe and winter in Africa, due to their recent decline and the series of linked conservation issues they face[7], the techniques developed here could be applied to any taxa for which sufficient occurrence data are available. Due to the rise in citizen science data collection[31], and the use of new recording devices like acoustic recorders[32], there is potential to extend this work to other taxa and regions.

Although the 2020 CBD target of 17% of land protected by 2020 has been achieved in Europe, existing protected areas are inadequate for meeting the dynamic protection needs of migratory species. Even 4 years after 2020, just over half our species (17/30) were adequately covered by protected areas for their entire time in Europe under our 17% target. None of our study species were adequately covered for their entire time in Europe under our 30% target (though Ring Ouzel was sufficiently represented for all but 1 week). Though the CBD targets were not designed to apply to species ranges, all of our selected species are widespread with a range over 250,000 km² for the majority of their time in Europe (including passage), and therefore we would expect the percentage cover of their ranges by protected areas to approximate that of the land surface. The mismatch between the proportion of land which has been protected and the proportion of a species' summed occurrence that falls within a protected area can arise from biases in the locations of protected areas. For example, protected areas are disproportionately located in places that are unsuitable for agriculture and urban development, rather than necessarily being sited in the most biodiverse areas[33–35]. Despite many papers highlighting the issues of area-based targets for protected areas[34–36] and despite international agreement in the 2020 CBD targets that protected areas should be sited in areas of particular importance for biodiversity and ecosystem services[15], the majority of new protected areas since 2004 are still sited on low-cost land with little agricultural value and, if anything, this trend is intensifying through time[35]. This practice can still benefit threatened species most associated with unproductive landscapes. For example, a surprisingly high proportion of Ring Ouzel's summed occurrence was covered by protected areas (Fig. 1), due to its alpine breeding habitat which is largely unsuitable for intensive farming and urbanisation[33,37]. In general, though, strategically siting protected areas in regions of higher biodiversity or higher ecosystem services, or targeting unrepresented and threatened species, would improve the effectiveness of protected areas[24,35] and could be much more economical in the long-term[38]. Our results indicate a valuable opportunity for future protected area planning in Europe to explicitly address the dynamic protection needs of threatened migratory species[7], which, this study suggests, could benefit significantly from such areas.

Of the species we assessed, the Ortolan Bunting was the least well covered by the protected area network by a considerable margin. Its western flyway population is threatened by hunting and agricultural intensification on the breeding grounds, and flyway-scale conservation measures are required[39]. Eurasian Hoopoe *Upupa epops* was the next least well covered species (Fig. 1) and both species are primarily agricultural specialists[40,41], a habitat that the protected area network does not represent well[33,35,37]. Alternative conservation approaches, such as a land sharing approach integrated into farming[42,43], may hold more promise for these species.

The weekly assessments of ranges and protected area intersections we showcase here can help to pinpoint the times of year when species are most at risk from lack of cover by protected areas. For example, using approximate broad seasonal categorisations and our 17% target, we can see that European Nightjar and Common Wheatear *Oenanthe oenanthe* are inadequately covered on spring passage, Pied Flycatcher *Ficedula hypoleuca*, Tree Pipit *Anthus trivialis*, and Sedge Warbler *Acrocephalus schoenobaenus* are inadequately covered during their breeding period and Common Redstart *Phoenicurus phoenicurus*, Lesser Whitethroat *Sylvia curruca* and Willow Warbler *Phylloscopus trochilus* are inadequately covered during autumn passage. This information can be used to make effective and targeted conservation decisions. For example, if a species is not declining, and only inadequately covered by protected areas for a few isolated weeks it might be considered a low priority for protection, whereas a threatened species which is inadequately covered for the majority of its time in Europe might be considered high priority. Overlap with protected areas seems lower during autumn passage than at other times of year (though not significantly lower than in the breeding season), possibly suggesting a lack of protected area cover in autumn migratory corridors, when mortality may be greatest[44]. Of course, a species may only be transiently in autumn passage areas, and therefore, temporary measures–such as seasonal hunting bans[45] or grazing restrictions–could be effective for passage areas.

We found that species with a more positive population trend tended to have a higher portion of their summed occurrence in protected areas. This fits with findings from (Barnes et al.[46]) that bird population trends in Europe tended to be more positive in protected areas. However, findings from other studies are conflicting[11,12,47] and, in general, the effectiveness of protected areas depends on location, protection level, management activity, taxa and species[46]. One of the main drivers of change in European birds has been agricultural intensification[48,49], providing a plausible link to the benefits of protected area status which is more likely to maintain areas of natural and semi-natural habitat[35]. The fact that we found farmland species had a lower proportion of their range in protected areas, on average, compared to the 'other' species supports this theory, although we were unable to differentiate species more precisely by how much they specialise in farmland habitats. Revising the Annex 1 of the Birds Directive to include more migratory and farmland species would help to make these species a higher priority when designating protected areas. Declines have also been high in long-distance migratory birds compared to short-distance migrants[50], but when we removed the three short-distance migrants from the analysis we still found a significant positive correlation between trend and proportion of summed occurrence in a protected area ($p = 0.0026$), indicating that the link between protected area coverage and trend is not due to a spurious correlation with migratory distance.

There are some important caveats and limitations to consider when interpreting these results. Here we compared protected area cover to range-adjusted targets for individual species, based on the 2020 and 2030 CBD targets, and highlight the species and seasons when coverage is greatest or weakest. Due to uncertainties in our estimates of protected area coverage and the protection targets–such as the assumptions about coverage, how far the benefits of protection extend beyond protected areas, and the lack of a strong biological foundation for these policy targets–we cannot be sure if achieving the target percentages in our study will be enough to guarantee long-term survival. However, the positive association between protected area coverage and long-term population trends provides important new evidence that extending protected area coverage in Europe is likely to benefit declining long-distance migratory birds, assuming the past is a good predictor of the future. Further research is needed to develop robust target setting approaches that are grounded in the life history, threat status and traits of species. For example, Taylor et al.[51] use

targets based on population viability analysis to explicitly account for the long-term viability. To develop such targets, some important considerations would include (i) how much a protected area and its management improves the persistence of a given species[11], (ii) how large a protected area needs to be to provide such benefits[52], and (iii) the extent and magnitude of protected area benefits to the immediately surrounding landscape[53]. More information is also needed on which species will benefit most from protected areas and which can benefit from changes to the wider landscape, such as agricultural environment policies[7].

Our study species are all passerine or near passerine Afro-Palearctic migrants; hence, our findings may not be generalisable to other species groups, such as waders or waterfowl, many of which are dependent on different habitats with different levels of protection. Our models are based on climate, habitat and bird distribution data collected between 2010 and 2019. Climate and habitat are changing rapidly[54] and this will likely change distributions and timing of migration for many species[55,56] which, in turn, may alter how well the current protected area network covers these species. A comparison of our main results with data from two exemplar years (Supplementary Methods 1, Supplementary Figs. 1, 2), suggests our conclusions have low sensitivity to annual variability. But, accounting for the effects of additional warming may still be important to consider when assessing how best to optimise the location of new protected areas for migrants[57].

Even when species are considered adequately represented by protected areas, protected areas vary substantially in the level of habitat management and protection present[11]. For instance, the level of protection and management for protected areas in IUCN categories I to IV is substantially higher than for categories V and VI[29,30]. The cover of protected areas for all species and weeks would decline substantially if we omitted protected areas in IUCN categories V and VI from our analyses (Fig. 1b). This is despite the fact that IUCN designations are missing for over 50% of the protected areas. Considering the difference in protection conveyed by protected areas of different IUCN categories, infilling missing IUCN categories should be seen as a priority. The 2020 and 2030 CBD targets also stipulated that the 17% or 30% protected area cover should be conserved through effective management, but that was not something we had the data to assess in this study. However, the methods discussed here could also be applied to more specific cases and thereby, at least partially, circumvent this issue of unknown protection level. For example, one could assess the adequacy of the European Union's (EU) Natura 2000 network in conserving terrestrial migratory species and ensure relevant stakeholders are engaged throughout the process to enable this work to be used in EU policy as part of the EU biodiversity strategy for 2030[17].

We have focused on a set of species, but we have not assessed protected area coverage for different populations within a species. To an extent, our focus on the protection of weekly distributions will help to highlight gaps in coverage for different stages of a species' lifecycle. However, it is possible that the area protected could be biased towards one population, leaving another mostly or entirely unprotected. It would be possible to extend this work for specific (sub)species (where data are available), using ringing and tracking data to determine the locations of the different populations throughout their time in Europe and thereby assess protected area coverage for populations. Here, we have shown examples of what is possible using our modelling framework but determining which approach is most suitable will depend on statutory aims and the decision should involve input from policy makers, conservation bodies and experts in the field.

The unstructured or semi-structured nature of the occurrence data means that we are estimating the distribution of probability of occurrence, not abundance. Although a higher abundance generally does translate to a higher probability of occurrence, the relationship between occurrence and abundance is non-linear and can vary between species and over space and time[58]. Comparisons of abundance and occurrence data have found that though these results will not be identical, especially at a fine spatial scale, broadly the patterns are qualitatively very similar[58]. Therefore, we are confident that these models can still give us a good indication of how well species are covered by protected areas. Although our models provide a good characterisation of migratory bird distributions across Europe, we would recommend using direct observation to guide future protected area placement rather than model predictions. This is especially true in the migrating period, as STEM predictive performance has been shown to be slightly lower (though still better than or equal to other modelling methods) during periods of active movement compared to periods when the population is relatively static[27].

We have limited ability to account for detectability effects in these data due to a lack of information on survey area, survey duration, experience of the observer, time of day and survey conditions (visibility/weather) for the vast majority of records. There will likely be detectability differences between species, therefore, we do not attempt to compare probability of occurrence between species. There will also be seasonal detectability differences within a species, for example, when males are singing to attract mates versus when females are staying cryptic to incubate eggs or during moult. Though sex-based differences in migratory timing are typically 0–4 days[59] and we therefore do not anticipate this variability substantively influencing the weekly distribution patterns we found here. However, crucially, none of these detectability issues should have a strong influence on the proportion of a species summed occurrence in a protected area, as none of these detectability issues are likely to show a strong bias towards or away from protected areas. One confirmation of this is that European Nightjar and Common Wheatear both have the lowest proportion of their summed occurrence covered by protected areas in spring, despite their contrasting ecologies and detectabilites.

Lastly, for 10 km squares which are only partly covered by a protected area, we cannot know how much of the species' summed occurrence is actually in the protected area. As we do not have information on species occurrence at a scale finer than 10 km squares, we make the assumption that, on average, over the course of a week, the species is equally distributed throughout the square[60]. We thereby aim to avoid the commission and omission errors caused by arbitrary thresholds for defining how much protected area cover counts as protected[60–62]. We tested the impact of this assumption by also using an alternative approach categorising 10 km squares as protected based on either 10% or 50% protected area coverage within them (Supplementary Methods 2, Supplementary Fig. 3). Although this shows that the absolute values of coverage vary depending on the assumptions made, the species and temporal patterns remain and are robust to these assumptions. Where a species is distributed in a square will depend on many things including time of day, date, habitat preferences (which will often change with time of day and date) and other individuals or species for reasons of competition, aggregation or predation[63]. Depending on its habitat and management, a protected area may be more or less attractive to the migratory bird species included here. However, as far as we know, this is the finest spatial-temporal resolution attempt to measure avian protected area coverage at this scale, with other papers using much coarser range maps e.g. [19] which will be far less accurate even for stationary periods and very inaccurate for movement phases. As yet, bird occurrence data at a grain finer than 10 km square does not exist at the same spatial-temporal scale and extent as used here, but more detailed finer scale measures of protected area cover could be achieved for some European countries e.g. [12], where this data is available from national structured surveys.

To conclude, we introduce an approach for assessing protected area coverage for mobile species. This approach could be refined to focus on particular types of protected area, or particular species, or

expanded to assess a larger range of species and taxa where data are sufficient, though we have discussed certain caveats that need to be taken into account when making policy decisions. Migratory species need carefully designed protection measures to address their rapid declines[7], but their dynamic distributions are rarely considered when setting policy responses[21]. Migratory species' distributions can change drastically through the weeks of the year, and a species apparently adequately represented by protected areas at one time of the year may not be well represented at other times of the year. Our analysis found a bias in protected area coverage where widespread species had a lower range coverage by protected areas than the wider landscape. Given the association between population trend and the proportion of species occurrences in protected areas, addressing these gaps for migratory land birds, potentially as part of a global initiative to increase protected area coverage to 30% by 2030, may also make a significant contribution to the future conservation of these species, many of which are declining in Europe.

## Methods

### Avian occurrence

**Underlying data**. Bird occurrence data for 30 passerine or near-passerine Afro-palearctic migrants (Supplementary Table 2 for species list including scientific names) were obtained from EuroBirdPortal (EBP; https://eurobirdportal.org), a European Bird Census Council (EBCC) project, with 81 partner institutions from 29 different European countries, which collates bird records from 18 different portals and combines them into a single online dataset (Portals listed in Supplementary Methods 3). These 30 species were chosen as they were all the passerine and near passerine species included in EBP at the time. We used observations collected between 2010 and 2019 at a 10 km × 10 km European grid square resolution (EPSG:3035 coordinate system), or 10 km square, for each day. We excluded EBP records from remote island groups and countries with very little data as we could not make reliable models for these regions (0.065%). We excluded data from organised surveys explicitly targeting non-target species (e.g. seawatches and raptor surveys, 0.7%). A subset of older records (29%) were removed from the dataset, because they were aggregated at the weekly level for each 10-km square, preventing us being able to determine species presence or absence in a list. EBP data come in two forms: complete list data and casual records. Complete lists arise where a birdwatcher recorded all the birds they saw or heard during a site visit. Casual records are ad hoc records made by birdwatchers, in which an unknown proportion of the species detected are recorded. The only measure of effort available for all EBP data is the total number of records. For complete lists, this is the total number of species in the list. For casual records, the record total is the total number of different combinations of observer and species recorded on that particular day in that particular 10-km square. We found that overall coverage of habitats was comprehensive, but urban habitat was overrepresented while forest habitats were represented approximately half as frequently as expected (Supplementary Table 3).

**Environmental variables**. Environmental data for our models, which included habitat, climate and elevation data stored in a series of rasters, were extracted for each 10 km by 10 km grid square included within the EBP dataset (EPSG:3035). If the raster data was in a different spatial projection to our 10 km square grid, we transformed our grid to the spatial projection of the raster for extraction purposes to avoid changing the underlying values of the raster, and then matched the extracted data back to the original 10 km square grid with the EPSG:3035 projection. We used exactextract from the R *exactextractr* package (v 0.8.2)[64] to extract the data. To create a full prediction data set we used the R sf package (v1.0.7)[65] to make a 10 km by 10 km square grid in the same projection as the EBP 10 km grid system and extracted the data using the same method as above.

For the model to determine pseudo-complete lists from casual records (see below), we used Copernicus Global Land Service 2015 Land Cover data at a 100 m by 100 m resolution (https://land.copernicus.eu/en/products/global-dynamic-land-cover). We extracted the mean percentage cover per 10 km by 10 km EBP square of each of the following key habitat variables: crops, grass, bare ground, seasonal water, trees, shrubs, urban, permanent water, moss and lichen. Elevation data came from Copernicus Land Monitoring Service (https://land.copernicus.eu/imagery-in-situ/eu-dem/eu-dem-v1.1), at a 25 m resolution. The mean elevation per 10 km by 10 km square was extracted. For this model we wanted an approximation of complete list length based on time of year and habitat that could be generalised to apply to both areas with casual records and complete lists and areas with just casual records. Therefore, we used a reduce set of more general variables that would broadly capture the main variation in the habitat and time of year.

For the Spatiotemporal Exploratory models (STEM, see below) we used a larger variety of more detailed habitat and climate data as we were trying to specifically describe the exact patterns in occurrence present at each place and time and were not aiming to predict outside the range of the raw data. The variables we used are described in Supplementary Data 1 and were originally derived for the European Breeding Bird Atlas (as described in Milanesi et al.[66]). With such a large area to model (112,015 10 km squares) we were limited in how many variables we could include by the memory limits on the JASMIN super computer, therefore, we were not able to include each habitat variable twice, once at the local 10 km square scale and once at the larger landscape scale, summing values for the nine 10 km squares surrounding each 10 km square. However, the landscape scale variables were highly correlated to the local scale ($r > 0.7$) and therefore would not have given much additional information.

**Generating pseudo-complete lists from casual records**. To generate pseudo-complete lists from casual record data, we subsetted the EBP data to include only complete lists ($n = 3,054,963$) and modelled the total list length against the coordinates of the complete list, month of the year, elevation and habitat present (mean percentage cover of crops, grass, bare ground, seasonal water, permanent water, trees, shrubs, urban, moss and lichen) using a random forest model. This model was then used to predict expected list length for every 10 km square in Europe. A set of casual records for a particular 10 km square and day was included as a pseudo-complete list if it had equal to or more than the predicted complete list length for that square/month combination. Following this approach, we included 1,851,697 pseudo-complete lists and 3,022,531 complete lists (for distribution see Supplementary Fig. 4a).

### Spatiotemporal exploratory models

To obtain weekly predictions for each species' probability of occurrence for every 10 km square in our study area, and thereby determine how a species was distributed in relation to protected areas, we used STEM[27,67]. We modelled species occurrence (inferred from presence or absence in a complete list or pseudo-complete list) against easting, northing, year, ordinal day of year, topography, habitat (22 variables including soil type, forest type and canopy height, type of vegetation cover, diversity and Normalised Difference Vegetation Index) and six climate variables (including temperature, rainfall and evapotranspiration) (see Supplementary Data 1 for a complete list of variables and sources). The EBP data will likely only include birds actively using the habitat as our study species generally migrate at night[68], and other studies have found habitat associations with migrants in active migration[69,70].

STEM is a predictive ensemble model for non-stationary spatio-temporal processes. To run the STEMs, first, the study area was split into an evenly distributed spatial and temporal grid of stixels (or

spatiotemporal blocks)[27,67]. An independent 'base' model was fitted to the data within each stixel. This process was then repeated for a specified number (*n*) of randomly positioned grid partitions using the same stixel dimensions, resulting in an ensemble of partially overlapping base models[58]. Within each stixel, the relationships between species occurrence and the predictor variables are assumed to be stationary but, between stixels, these relationships can vary. For example, a species may be differently influenced by habitat depending on whether it is migrating, breeding or moulting and, therefore, for the same area, the predictor/response relationships may vary for different temporal windows. Predictions for a particular location and time were calculated by averaging over all *n* base model predictions that contained the target location and date; the variance for each prediction was the variance between the base model predictions.

As STEM is very computationally intensive, all analyses were run on the supercomputer high memory node (JASMIN; https://www.ceda.ac.uk/services/jasmin/). Despite this, we still had to compromise between model predictive accuracy and computational feasibility. We used Generalised Boosting Machines as our base model, with 1000 trees, 0.95 bag fraction and interaction depth of 3. The stixel width was set to 500 km by 500 km by 40 days, having also trialled stixel widths of 100 km, 200 km, 300 km, 400 km and 600 km and 20 days. The selected stixel width was a compromise between predictive accuracy, computational feasibility and coverage (smaller stixels led to more gaps in coverage due to insufficient data per stixel). To validate the predictive power of our models, we used 5-fold cross validation and split the data into folds using environmental blocking to ensure the training and test data were independent. The AUC statistic was used to assess the predictive accuracy of our models. We set the minimum data requirement for each stixel to 30 lists and *n* (the number of base models to fit) to 10. Although preliminary analysis attempted model fitting with *n* = 25, it was not possible to run 5-fold cross validation for all species with *n* = 25 due to memory constraints.

Lastly, we made predictions of species occurrence for each 10 km square and week (week 1 = 1st–7th January) based on the final STEM for each species using our full prediction dataset detailed in Supplementary Data. We specified the total list length as 25, year as 2018 (see Supplementary Methods 1) and the data type as complete list.

In STEM, a very low number of occurrences for the modelled species in a particular area of space and time leads to low ensemble support, as the number of base models which can be fitted is restricted by the minimum sample size. To remove this effect, predicted occurrences below 0.01 (predicted occurrence was between 0–1) were set to zero, as predictions below this threshold represented a very small number of occurrences that did not represent the population as a whole and would lead to overpredictions of species ranges[71].

### Protected area coverage

**Protected area data.** We downloaded protected area data from Protectedplanet.net[72]. We cleaned the protected area data using the wdpar R package[73] following best practices (https://www.protectedplanet.net/en/resources/calculating-protected-area-coverage). This removed (1) protected areas that are not currently implemented (keeping Designated, Inscribed or Established), (2) United Nations Educational, Scientific and Cultural Organization (UNESCO) Biosphere Reserves (0.07%) (as these contain large areas not protected[74]), (3) point locations without an area specified (2.35%), and also repaired any invalid geometries as needed (e.g. due to self-intersecting features). Points were converted into buffered circular areas using the reported area of the associated protected area. We also removed areas representing Other Effective Conservation Measures as these are not officially designated protected areas, and all protected areas whose designation was listed as not assigned, not reported or not defined (0.35%). The dataset was then transformed into the Lambert Azimuthal Equal Area Europe projection and dissolved to remove overlaps. We dissolved the dataset by IUCN designations progressively from categories Ia to VI, and then sites with no assigned IUCN category, so high protection designations replaced lower protected area designations in places where these overlapped. Next, we intersected the 10 km × 10 km grid used for the EBP observational data (see above) with the protected areas shapefile, to get the proportion cover of protected areas for each 10 km square (see Supplementary Fig. 4b).

To calculate the proportion of a species' summed occurrence (i.e. total probability of occurrence summed across its entire range) that was in a protected area, we estimated the percentage of a species' total weekly summed occurrence in each 10 km square of the study area. We then multiplied this value by the proportion of each 10 km square covered by a protected area, for example if a square contained 10% of a species' summed occurrence and half of it was in a protected area then we assume that 5% of the species' summed occurrence was protected in this square (following Araújo et al.[60]). This approach avoids the likely errors in commission and omission inherent in arbitrary thresholds for determining whether a 10 km square is protected or not[60–62]. To explore the variation in percentages of a species' weekly range that would be considered protected under some of the commonly used thresholds, we carried out an additional supplementary analysis (Supplementary Methods 2). We also explored how temporal changes in species distributions due to changes in arrival timing between years might affect the weekly percentage protected area cover for a species (Supplementary Methods 1), though we expect this to be a minor effect[75].

**Representation by protected areas.** We assessed two protection target setting approaches, based on the 2020 and 2030 CBD targets, but adjusted by a species' range following Rodrigues et al.[23]. We used loglinear interpolation between two thresholds, where 100% of a species is protected if its range is below 1000 km² and 17% (2020 CBD) or 30% (2030 CBD) protected if its range is above 250,000 km² see ref. 19,23. By basing the protection target on weekly range size, we aim to highlight vulnerable weeks when a high proportion of a species' population is concentrated into a small area. This would ensure that a species' weekly protection target was related to their European range size at any given time, with high protection for smaller ranges[19,23].

To avoid underestimating the effectiveness of protected areas in Europe when a species is mostly in the African stage of its migratory cycle, we capped species' weekly distributions to span from the earliest to the latest week that the species' summed occurrence was equal to or greater than 25% of the maximum summed occurrence. Various cut-offs were trialled (15, 20, 25, 30, 35) before settling on 25%, and assessed using visual examination of the weekly distribution maps, the potential cut off dates and knowledge of species' typical breeding seasons[40]. Summed occurrence is influenced both by the abundance of a species in an area and its detectability. Consequently, occurrence varies through the breeding season as a result of varying detectability (e.g. singing periods versus incubation and moulting). Summed occurrence cut-off thresholds higher than 25% resulted in some species being considered absent from Europe during the middle of the breeding season which is manifestly not the case.

To determine whether protection level differed depending on the time of year or range size, we used a Linear Mixed Model. The model used the weekly percentage of a given species' summed occurrence in a protected area as the dependent variable, and the species' weekly range size and time of year as predictor variables. 'Time of year' was a categorical variable denoting spring passage [weeks 5–18], breeding season [weeks 19–31] and autumn passage [weeks 32–48]. Note that there is much overlap in breeding and migration periods across Europe and these broad categorisations are not meant to be absolute. An intercept only random effect of species was also included in the model to account for repeat measures across weeks for each species.

Previous studies have found that protected areas tend to occur in more natural landscapes rather than in farmland and urban areas[33,35], therefore we also assessed whether the mean percentage of a species summed occurrence in a protected area (over all weeks) was related to species main European habitat (as defined by PECMBS[41] and for nightjar Sharps et al.[76]), in a linear regression. Due to our small sample size of 30 species, we used a relatively coarse habitat classification with three categories (farmland- which includes grasslands as well as agriculture lands, forest, and other)[77]. This coarse classification does not capture the extent to which species may specialise or generalise within these categories.

Lastly, to assess whether species' 30-year Pan-European Common Bird Monitoring Scheme (PECBMS) trends (https://pecbms.info/trends-and-indicators/species-trends/) were associated with range size or protected area coverage, we used linear regression (LM) to fit the PECBMS long-term trend as a function of range size and protected area coverage. To do this, we averaged range size and proportion of summed occurrence in a protected area over all weeks for each species. The PECBMS long-term trend was converted to a $\log_{10}$ ratio to ensure that the same geometric change is symmetrical around zero (i.e. a 50% decrease was equivalent to a 50% increase). Sand Martin *Riparia riparia* and European Nightjar were not included in these latter models, since PECBMS trends were not available for these species. We also tried including habitat (as described above), migratory distance (long distance migrant = winters mainly south of the Sahara, short-distance migrant = winters mainly north of the Sahara)[7,78] and log body mass[79] (Source Data), in the model to ensure any apparent effect of protected area was not a surrogate for population effects of habitat, migration distance or body mass[50]. Since the sample size was only 28 for this analysis, we could only include three variables at a time to avoid over parametrising the model[77]. Additionally, because only three of our species were short-distance migrants and the rest were all long-distance migrants, and therefore our ability to account for the effect of migration distance may be limited, we also tried removing these species and repeating the analysis to ensure that our results were robust.

All analyses were conducted using the R statistical computing environment (version 4.2.0)[80] and packages used are listed in Supplementary Methods 4. Model assumptions for the LMM and LM were tested via plotting residuals, dependent variables and covariates and found to be sufficiently met. All statistical tests used the alpha value of 0.05 and all model parameter estimates are presented in the results accompanied by the standard error after the ± symbol.

### Reporting summary

Further information on research design is available in the Nature Portfolio Reporting Summary linked to this article.

## Data availability

The raw bird occurrence data analysed in this study have been deposited in the EuroBirdPortal database under the data request id: BTO_03_2020. The raw bird occurrence data are available under restricted access to ensure proper use and knowledge of the data. The data belongs to EuroBirdPortal partners, access can be obtained (for purposes of replicating the analysis) by emailing Gabriel Gargallo on anella@ornitologia.org and Verena Keller on verena.keller@vogelwarte.ch, requests will be responded to within a week and access will be granted for as long as needed. The processed data (predicted occurrences for each species at a 10 km square resolution, the percentage cover of each 10 km square by protected areas, and animated and stationary weekly distribution maps) are available at Zenodo repository (https://doi.org/10.5281/zenodo.10960419). The source data for all graphs generated in this study are provided in the Source Data file. The protected area data

used in this study are available in the Protected Planet database from www.protectedplanet.net.

## Code availability

All code was written in R using open-source packages and functions and can be accessed Zenodo repository https://doi.org/10.5281/zenodo.10960419.

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

## Acknowledgements

We thank all people who contribute observations to the different online bird recording portals operating in Europe and to the EuroBirdPortal project partners for providing the data. This is a contribution to the EU funded project 101104367 (LIFE22-PLP-ES-EBP reinforcement, L.B., G.G., H.S., R.P.B.F.). We acknowledge the E-OBS dataset from the EU-FP6 project UERRA and the data providers in the ECA & D project (https://www.ecad.eu). Thanks to The Science & Technology Facilities Council and the Natural Environment Research Council for free use of the JAS-MIN super-computer. J.O.H. was supported by Environment and Climate Change Canada (ECCC) and Nature Conservancy of Canada (NCC). This project was funded by NERC grants to the British Trust for Ornithology and the University of Durham (NE/T001070/1, C.H., S.G.W., P.A.S. and NE/T001038/1, J.A.B., S.R.B., C.M.H. and J.W.P.H.), and additionally, generously supported by a gift in Will from Michael Welch, for which we are extremely grateful.

## Author contributions

J.A.B, S.R.B., J.W.P.H., S.G.W., P.A.S. and C.H conceived the study. J.A.B undertook the analysis and wrote the first draft with extensive contributions from J.W.P.H, S.R.B., R.A.F., J.O.H. G.G and R.B.P.F. and S.R.B led the EuroBirdPortal data gathering and H.S. extracted the environmental data for the STEM analyses. D.F. advised on the STEM analyses. J.A.B, S.R.B., J.W.P.H., S.G.W., P.A.S., C.H, C.M.H, H.S., J.O.H., G.G., R.B.P.F., R.A.F., D.F. and L.B. provided useful comments and feedback on draft versions of the paper and revisions. All the authors contributed substantially to the final paper.

## Competing interests
The authors declare no competing interests.
