## [Transparent Peer Review file · Nature Communications]

Expanding protected area coverage for migratory birds could improve long-term population trends

Corresponding Author: Dr Jennifer Border

Version 0:

Reviewer comments:

Reviewer #1

(Remarks to the Author)

General comments

We found the topic of this contribution important and timely. Overall the manuscript is very well-written and executed, and we have but few points that could be clarified and acknowledged.

You address some of the limitation of your work in the discussion. However, there may be others that may want to consider: Your citizen science data may entail spatial Biases that may affect your results. For example, if you may have less data from farmlands. Perhaps explore the proportion of records across habitats when compared to their spatial prevalence?

Your data is based on a decade of observations. While migrating birds may have strong philopatry, the specific timing of their migration may be highly dependent on short-term climatic conditions. These changes may be exacerbated in the era of climate-change we live in. We think this should be acknowledged and potentially also quantified – for example as some variance measures for your weekly temporal predictions (across the years).

In the original paper that presented STEM Fink et al. (2010) highlighted that their model performed better during the periods where the birds were mostly stationary rather than actively migrating. Perhaps worthwhile mentioning this.

The differential migration timings of bird sexes (Protoandry etc.) has been shown to occur in dozens of species (Lehikoinen et al. 2017). This may have ramifications on your results, though if the differences in timings are within your weekly temporal resolution – this may be minor. Nevertheless, this may be worth mentioning as a caveat.

It would also be important to stress in your limitation that your exploration of predominantly passerines may be misleading for other groups of migrants that may depend on other habitats such as waterfowl and waders.

We also think you need to acknowledge that a. the habitat classifications you use are very broad, and b. some of the species you explore are more habitat specialists or generalists – and this too, may explain some of the trends you find across species.

Specific comments

Perhaps in the abstract used a more common term such as “occurrence”, rather than “summed occurrence” as this term may be confusing without further explanation.

Also, perhaps in the abstract if you mention the “EuroBirdPortal” – explain that it is a conglomerate of the most used bird occurrence atlassing apps, as this platform may not be familiar to many readers. Surely give some more details on this source in the introduction.

Line 82 – might as well say Afro-Palearctic here and throughout, since this is the term you use in the discussion – and the more commonly used term for this migration pathway.

Line 92 – I think this is your first use of the common name, so please include the scientific name for nightjar here.

You refer to some of the species you explore as “farmland” species probably due to the CBD naming. However, farmland implies their preferred habitat is farm, whereas it is actually grassland or natural fields, and farms are simply the closest they can often get and perhaps this needs acknowledgment.

Bibliography

Fink, D., Hochachka, W.M., Zuckerberg, B., Winkler, D.W., Shaby, B., Munson, M.A., Hooker, G., Riedewald, M., Sheldon, D., Kelling, S., 2010. Spatiotemporal exploratory models for broad-scale survey data. *Ecological Applications* 20, 2131-2147.

Lehikoinen, A., Santaharju, J., Anders, M., 2017. Sex-specific timing of autumn migration in birds: the role of sexual size dimorphism, migration distance and differences in breeding investment. *Ornis Fennica* 94

Reviewer #2

(Remarks to the Author)

This paper uses an amazing set of citizen science data collected via the EuroBirdPortal in a novel way to model migrant bird species occurrence to assess how their weekly distributions overlap with protected areas (PAs) in Europe. That is an interesting and very worthy analysis. They also show that a species' summed occurrence in a PA was positively correlated with long-term population trends, after correcting for other factors – suggesting PA were acting positively. However, the focus of the paper is a comparison between migrant bird overlaps with PAs against the rather dated 2010 CBD Aichi biodiversity target for PAs coverage set at 17% and subsequently with the 2022 CBD KMGBF target of 30%, which is rather underplayed. The link between these two different things is unclear on a number of levels. The CBD targets relate to recommended percentages of the Earth that should be protected by signatories, so the CBD targets do not relate to a recommendation of the quantity of any species range that should be protected. The leap from one to the other in the paper is hard to follow or understand. Expressing the results as a difference from the 2020 and 2030 target thresholds is odd. An independent and different reference comparison with a more biological basis might make more sense, or just some fixed values. I wonder if that might be available from related literature. Furthermore, the Aichi target from 2010 is no longer policy relevant so any contemporary analysis of that target feels outdated. For this reason, the primary aim and focus of this paper is questionable, at least, not properly explained or justified. My only other query is that the bird records are at a 10km scale so, as the authors say, they do not know where in a 10km a species actually occurs and so whether that occurrence was in, or outside a PA. This is a limitation that needs more careful discussion and is a major caveat.

Specific comments

Line 49. Surprising to see no reference to M Somveille's leading work on bird migration here in this introductory paragraph.

Line 55. The comment: "many migratory species have been declining over recent decades, due to human activities such as habitat destruction, creation of barriers, over-exploitation and climate change" needs to be supported by more evidence and be more circumspect tone. Some of these are hypothesised and suggested, not proven causal or evidenced links.

Line 63. What is referred to as the "2010 United Nations Biodiversity Conference" was the 10th Conference of the Parties to the Convention on Biological Diversity held in Nagoya, Japan. This is where the 'Aichi' biodiversity targets come from. Perhaps clearer to use that language, if such targets are used at all. The Convention on Migratory species and CBD COPs are related, but quite separate processes. Does the CMS advocate for a percentage of the migratory species range that should be protected? That seems more relevant to your analysis than CBD PA targets.

Line 67. When you say 'need to be adequately protected' what does that mean in a substantive sense? Has other work provided a guide or made suggestions? Theoretically, how much of a passerine or near passerine species' migrants' species range should ideally be covered by PAs? And on what basis? Presumably that might be different for other kinds of migratory birds, such as congregatory waterbirds or seabirds? Do we know the answer to these questions? If not, how might it be reasonably estimated to inform your work?

Line 84. The ambition "By assessing protection on a weekly basis, we aim to account for migratory species' mobility to gain an accurate picture of the level of protection currently conferred and to uncover any gaps in the current protected area" is extremely laudable and would be new.

Line 98. There is live discussion as to whether the inclusive PA categories (V and VI) should be counted within 'PAs' as they may confer no conservation benefit as cultural living landscapes. Is there value in repeating your analysis without those categories? It is regrettable that >50% of PAs have no reported category, that makes understanding their impact even more difficult. It would be important to flag this issue in the discussion so that future research in this area might be improved.

Line 138. You state: "We found that, currently, no species was adequately covered by protected areas under the 2030 CBD target throughout the European stage of their lifecycle" but for reasons given above, that statement as is, is misleading.

Line 276. Agreed: "Here, we introduce an approach to assessing protected area coverage for mobile species. This approach could be refined to focus on particular types of protected area, or particular species, or expanded to assess a larger range of species and taxa where data are sufficient, though we have discussed certain caveats that need to be taken into account when making policy decisions." There is genuine value and novelty in this analysis and data use, but I struggle with the current focus.

Line 284. You state: "Our analysis found serious deficiencies in protected area cover for the 30 species of 284 Afro-Palaearctic migrant we assessed, we assessed, under both the 2020 and 2030 CBD targets" again, this is very questionable.

Line 291. The quite novel methods are well described and carefully set out. However, I'd like to see some maps with the data coverage of the bird data and PAs to get a better sense of 'Europe' and better understand the analysis.

Line 387. The calculation of the proportion of a species' summed occurrence within a PA is the other part of this paper that needs some care, as stated by the authors, they do not know where in a 10km a species actually occurs and so whether that occurrence was in, or outside a PA. The approach taken is reasonable but this is somewhat inconsistent with the rather dogmatic and precise description of the results and their interpretation in the abstract and discussion. It is touched on in the section on Limitations but not spelt out and emphasised. Were other approaches explored to deal with this issue or at least estimate sensitivity?

Reviewer #3

(Remarks to the Author)

This is an original manuscript examining the use of space and the use of protected areas by migratory birds (mostly passerines) across their annual life cycle while they are in Europe. Hence, including the breeding and the migration periods. The authors use citizen science data to model the distribution using a probability of occurrence and species distribution models including habitat/land use variables and topography. The SDMs were done at 10km² resolution across Europe.

The overall idea and analysis is original and new but the rationale is not fully clear or is not coherent. The scale of the study is not sufficiently detailed. It would be good to see more details on how small PAs (<10km²) were considered in this study. It may be that within an 10km² the bird distributions are concentrated in PAs (the effect of small PAs needs to be considered/discussed). It is possible that some species may have distributions concentrated in PAs but this is not detectable at 10km² resolution. I acknowledge that the nature of the data, collected as part of citizen science programmes may not enable a finer scale analysis (e.g. some of the complete species lists used may have been collected over large areas) which does not enable the authors to work at a smaller spatial scales. This is a problem that needs to be addressed from the perspective of the PA network. How many areas are less than 10km² and what was the approach adopted for small PAs? It is possible that at a finer resolution (many protected areas are small and do not occupy a full 10km²) the species distribution may be more concentrated in PAs but the data and the analysis did not enable the authors to determine if this is the case.

1- The authors investigate the use of protected areas (PAs) which, at present, correspond to 20.1% of Europe, by 30 migratory passerine and non-passerine species. The majority of the species are widespread (ranges >250,000km² and have 20% of their distribution in PAs (corresponding to the PA area available) as expected. What is adequate protection?
2- PAs have a variety of criteria behind their classification, while some were classified to protect endangered bird species others may have been designated due to landscape features or habitats and may not be particularly effective at protecting migratory species. In most countries PAs have natural or semi-natural landscapes and will be effective at protecting the species that depend on those features, not the wider countryside species. The study explores if the current PA network adequately protects migratory species but the network was not built for this purpose. This should be better explained.
3- The Natura 2000 SPAs were classified to protect endangered bird species and many migratory species included in this study were not part of the priority list. The rationale for the study is not fully justified, why should the PA network covering 20% of the land protect more than 20% of the distribution of the migratory species, if this was the case it would indicate that migratory species were becoming concentrated in PAs disappearing from the wider landscape, which is not desirable. Overall the rationale of the paper is poorly explained, it is preferable to find that migratory species are still found across large areas rather than concentrated on PAs (having said this the scale of the study makes it difficult to conclude this with certainty).

The rationale for the study needs to be better explained.

4- The modelling procedure, done on a weekly basis, and including the migration period may include birds migrating and not using habitat, hence modelling habitat use during migration may not be fully appropriate. This should be justified.

5- It is interesting to note that during migration many species have very low representation in PAs which suggests that there are very few PAs in migratory corridors which could have been better discussed by the authors.

6- The authors find that species that have higher representation of their distributions in PAs are doing better (likely species benefitting from conservation actions and habitat management measures) than those that occur in the wider countryside. This is not new.

7- Eurasian Hoopoe and Oortolan Bunting seems to have less than expected distribution in PAs, across the whole time they spend in Europe, this suggests that the PA network is particularly poor for these species. This is an interesting result that could have been discussed in more detail.

8 - To achieve the 2030 CBD target we would need to classify 30% of the planet, and in this case we would expect to protect 30% of the distribution of these migratory species. It is not expected or desirable that the current 20% of Europe that is classified as PAs should have 30% of the distribution of these widespread species (it would mean these species would no longer be widespread across Europe).

Methodological issues

8- The choice of the 30 species included in the study is not justified.

9- What do you mean by adequately protected (e.g. line 158)? Would the authors expect that more than 17% (CBD2020 target) of the distribution should be included in PAs? This was not fully explained.

Reviewer #4

(Remarks to the Author)

Version 1:

Reviewer comments:

Reviewer #1

(Remarks to the Author)

We acknowledge the effort done by the authors of this contribution and are happy with its current status

Reviewer #2

(Remarks to the Author)

Very pleased to see this revised manuscript and the care taken in responding to my original comments and concerns. The revised manuscript is much improved and reads very well. I have no substantive comments but some suggestions and

observations below.

Incidentally, it did occur to me that you could use this analytical framework to specifically identify and map priority areas in Europe for area protection for migratory landbirds towards 30x30, in a more propositional way, perhaps that is your next paper.

Line 105 & 179. The species here are essentially terrestrial species, landbirds, so it would be good to make that clear upfront. You aren't including seabirds, waterbirds, raptors etc that readers might think of immediately as typical migratory, congregatory birds.

Line 202. Here and elsewhere, meeting an area target for PAs tells us nothing about site quality and management as you touch on later and isn't strictly the CBD target. The CBD targets have area, plus 'effectively conserved and managed....etc etc'. It would be important to stress the two parts to the CBD targets – more 'paper parks' are likely to be of little use to conservation.

Line 249 – Could the lower overlap with protected areas during autumn passage be a product of observer recording behaviour. All birders are keen to see and record the first spring X species, but I wonder by autumn whether recording enthusiasm is as good and departing birds are as well recorded? Any comment or data to check on this?

Line 178 – While the Discussion is good it is very long. It seems to me that some of the very detailed 'caveat' paragraphs here might be better placed in the supporting materials to allow a better flow for the reader.

Richard Gregory

Reviewer #3

(Remarks to the Author)

This manuscript makes use of a strong temporally and spatially explicit dataset. The authors use information from the EuroBird Portal containing distribution information for 30 migratory European African passerine and near-passerine species, on a weekly basis, to assess the time birds spend in protected areas across their life cycle. This is a robust 10-year dataset with weekly information for the 30 migrant species. The authors modelled the weekly distribution using habitat, climate and elevation information obtained at the 10km grid reference using spatiotemporal exploratory models.

The authors addressed most of my previous comments and I am happy with the new version of the manuscript.

A few last comments/queries:

The authors mention "For protected areas to be effective in conserving migratory species, key habitats and resources at all stages of their life cycles need to be adequately protected" this is true but the current natura 2000 network does not target many of the species included in this manuscript so we would not expect these species to be well represented. Revising the species in Annex I of the Birds directive and including more migratory and farmland species should be made a priority?

The authors also state in line 331: "However, the positive association between protected area coverage and long-term population trend provides important new evidence that extending protected area coverage in Europe is likely to also benefit declining long-distance migratory birds, assuming the past is a good predictor of the future". As mentioned earlier in the manuscript, many of the species that are declining are farmland birds (not targeted in Annex1 of the Birds directive so not prioritised for conservation in PAs). The declining species are associated with agricultural landscapes that suffered transformations due to the intensification of agriculture. Classifying agricultural land as protected areas may help these species due to increased adoption/use of agricultural environmental policies. Would more funds be available to farmers in these areas to prevent further intensification of farmland? Taking into account funds for conservation are limited, setting up priority areas for farmland/migratory birds would concentrate resources in targeted areas instead of promoting horizontal measures (e.g. wildflower edges or parcels) across wider landscapes. I realise you are not yet sure which approach should be used for which species (line 340) but the consequence of more protected areas, and protected areas for the species in this study suggests you are proposing the use of more resources in targeted areas rather than more horizontal measures across the landscape. If this is not your recommendation you should make it clear.

Minor comments:

Line 1003 lacks a T in The colours

Supplementary Methods 4 Environmental Variables:

What do you mean with the sentence: "For the model to model complete list length against on space, time and habitat, we used Copernicus Global Land Service 2015 Land Cover data at a 100 m by 100 m resolution"

Reviewer #4

(Remarks to the Author)

RESPONSE TO REVIEWERS' COMMENTS

Please note all lines numbers refer to the track changes version of the manuscript.

Review 1:

We found the topic of this contribution important and timely. Overall the manuscript is very well-written and executed, and we have but few points that could be clarified and acknowledged. You address some of the limitation of your work in the discussion. However, there may be others that may want to consider:

1) Your citizen science data may entail spatial Biases that may affect your results. For example, if you may have less data from farmlands. Perhaps explore the proportion of records across habitats when compared to their spatial prevalence?

We added Supplementary Table 3 to the supplementary information which shows the percentages of EuroBirdPortal records in the key habitats we included in our model, relative to the wider landscape. We included the following text referring to this in the methods, lines 478 to 480: *"We found that overall coverage of habitats was comprehensive, but urban habitat was overrepresented while forest habitats were represented approximately half as frequently as expected (Supplementary Table 3)".*

2) Your data is based on a decade of observations. While migrating birds may have strong philopatry, the specific timing of their migration may be highly dependent on short-term climatic conditions. These changes may be exacerbated in the era of climate-change we live in. We think this should be acknowledged and potentially also quantified – for example as some variance measures for your weekly temporal predictions (across the years).

This is an interesting point, particularly where these changes in migratory timing affect the percentage protected area cover per week for a species. We would expect these changes to be minimal as we only look at weeks when a species' summed occurrence is greater than or equal to 25% of its maximum summed occurrence. In doing so, we exclude the earliest and latest arrivers, which would be expected to vary more (Miles et al. 2018). We re-ran the STEM model predictions for all species for two example years (2011 and 2015) to better understand the range of variation due to year and presented the results in Supplementary Methods S1 (as the STEM models are memory intensive and have long run times, it was not feasible to re-run predictions for all 30 species in all 10 years). We chose those 2 years to complement the year used in the predictions in the paper (2018). This ensured we had effectively covered the range of years included in the study (2010-2019), but avoided predicting for the earliest and latest years, as predictions on the edge of the data range can be less reliable and our climate data is focused in the middle of our range of years (Supplementary Table 4). We compared the results for these two years with the results used in the paper, and found the change in percentage of protected area cover per week between years was less than 2% in 92% of cases, and at most 7%, which only occurred in 0.1% of cases. Differences between years were largest during passage periods. We added the following text to the discussion (Lines 346-353), and the output of our additional analyses to the supplementary materials (Supplementary Figures 1 & 2):

Our models are based on climate, habitat and bird distribution data collected between 2010 to 2019. Climate and habitat are changing rapidly³⁴ and this will likely change distributions and timing of migration for many species^{55,56} which, in turn, may alter how well the current protected area network covers these species. A comparison of our main results with data from two exemplar years

(Supplementary Methods S2, Supplementary Figures 1&2), suggests our conclusions have low sensitivity to annual variability. But, accounting for the effects of additional warming may still be important to consider when assessing how best to optimise the location of new protected areas for migrants⁵⁷.

3) In the original paper that presented STEM Fink et al. (2010) highlighted that their model performed better during the periods where the birds were mostly stationary rather than actively migrating. Perhaps worthwhile mentioning this.

We added the following text to the discussion (Lines 390- 395): “Although our models provide a good characterisation of migratory bird distribution across Europe, we would recommend using direct observation to guide future protected area placement rather than model predictions. This is especially true in the migrating period, as STEM predictive performance has been shown to be slightly lower (though still better than or equal to other modelling methods) during periods of active movement compared to periods where the population is relatively static²⁷.”

4) The differential migration timings of bird sexes (Protoandry etc.) has been shown to occur in dozens of species (Lehikoinen et al. 2017). This may have ramifications on your results, though if the differences in timings are within your weekly temporal resolution – this may be minor. Nevertheless, this may be worth mentioning as a caveat.

We added the following text to the discussion at lines 406-408: “Though sex-based differences in migratory timing are typically 0-4 days⁵⁹ and therefore we do not anticipate this variability substantively influencing the weekly distribution patterns we found here.

5) It would also be important to stress in your limitation that your exploration of predominantly passerines may be misleading for other groups of migrants that may depend on other habitats such as waterfowl and waders.

We added the sentence into the discussion (Lines 344-346): “Our study species are all passerine or near passerine Afro-Palearctic migrants; hence, our findings may not be generalizable to other species groups, such as waders or waterfowl, many of which are dependent on different habitats with different levels of protection.”

6) We also think you need to acknowledge that a. the habitat classifications you use are very broad, and b. some of the species you explore are more habitat specialists or generalists – and this too, may explain some of the trends you find across species.

We have now acknowledged this in the paper methods at lines 616-619: “Due to our small sample size of 30 species, we used a relatively coarse habitat classification with three categories (farmland- which includes grasslands as well as agriculture lands, forest, and other)⁷⁴. This coarse classification does not capture the extent to which species may specialise or generalise within these categories.

And in the discussion at lines 306-308: “The fact that we found farmland species had a lower proportion of their range in protected areas, on average, compared to the ‘other’ species supports this theory, although we were unable to differentiate species more precisely by how much they specialise within farmland habitats.”

Specific comments

7) Perhaps in the abstract used a more common term such as “occurrence”, rather than “summed occurrence” as this term may be confusing without further explanation.

Done, we used the word *distribution* instead.

8) Also, perhaps in the abstract if you mention the “EuroBirdPortal” – explain that it is a conglomerate of the most used bird occurrence atlasing apps, as this platform may not be familiar to many readers. Surely give some more details on this source in the introduction.

We now expand our description of EuroBirdPortal in the abstract, to explain that it is a ‘pan-European citizen science bird occurrence dataset’, to provide context for the wider readership. The tightly limited word count of the Abstract is not the best place to elaborate on the sources of data in EuroBirdPortal. Instead, we give this detailed information in the Methods (lines 464-468).

9) Line 82 – might as well say Afro-Palearctic here and throughout, since this is the term you use in the discussion – and the more commonly used term for this migration pathway.

Thanks, we missed this initially and have changed as requested.

10) Line 92 – I think this is your first use of the common name, so please include the scientific name for nightjar here.

Done, thanks we missed this.

11) You refer to some of the species you explore as “farmland” species probably due to the CBD naming. However, farmland implies their preferred habitat is farm, whereas it is actually grassland or natural fields, and farms are simply the closest they can often get and perhaps this needs acknowledgment.

We added the text to the methods at line 616: *Due to our small sample size of 30 species, we used a relatively coarse habitat classification with three categories (farmland- which includes grasslands as well as agriculture lands, forest, and other)*⁷⁴

Reviewer #2 (Remarks to the Author):

This paper uses an amazing set of citizen science data collected via the EuroBirdPortal in a novel way to model migrant bird species occurrence to assess how their weekly distributions overlap with protected areas (PAs) in Europe. That is an interesting and very worthy analysis. They also show that a species’ summed occurrence in a PA was positively correlated with long-term population trends, after correcting for other factors – suggesting PA were acting positively.

12) However, the focus of the paper is a comparison between migrant bird overlaps with PAs against the rather dated 2010 CBD Aichi biodiversity target for PAs coverage set at 17% and subsequently with the 2022 CBD KMGBF target of 30%, which is rather underplayed. The link between these two different things is unclear on a number of levels. The CBD targets relate to recommended percentages of the Earth that should be protected by signatories, so the CBD targets do not relate to a recommendation of the quantity of any species range that

should be protected. The leap from one to the other in the paper is hard to follow or understand. Expressing the results as a difference from the 2020 and 2030 target thresholds is odd. An independent and different reference comparison with a more biological basis might make more sense, or just some fixed values. I wonder if that might be available from related literature. Furthermore, the Aichi target from 2010 is no longer policy relevant so any contemporary analysis of that target feels outdated. For this reason, the primary aim and focus of this paper is questionable, at least, not properly explained or justified.

Apologies that this was not clear. The 2020 target was included as a base level to compare against. Our motivation was that we should have already reached this target by now and, if we haven't, that is worrying. The 2030 target is to be achieved by 2030; consequently, we wouldn't expect to be meeting this target yet, but it is interesting to ask how far we are from reaching it.

However, in response to these comments, we have shifted the focus of the paper towards the percentage of a species' summed occurrence that is protected rather than comparing this only to the 2020 and 2030 CBD targets; we have revised Figure 1 to reflect this. Also, see changes to the text throughout the paper. To ensure a more biological basis for our target setting, we use Rodrigues et al.'s 2004 target for comparison of adequately protected. This adjusts the relative area requiring protection based on a species' range. However, the upper limit of 10% land in a protected area used in that paper is arbitrary, and was chosen purely because that was the global level of protected area cover at the time (Rodrigues et al. 2004). Consequently, we changed the upper target to 17% or 30% to reflect the changes to protected area targets since 2004. In this way, following Rodrigues, we assume widespread species should be protected to the same extent as their likely occurrence in the wider countryside. This is now explained in the introduction at lines 94 to 139, the methods at lines 579 to 586 and the discussion at lines 237 to 240 and 325 to 342. Also please see responses to comments 17, 18, 21, 23, 27, 28, 29, 34 & 36.

13) My only other query is that the bird records are at a 10km scale so, as the authors say, they do not know where in a 10km a species actually occurs and so whether that occurrence was in, or outside a PA. This is a limitation that needs more careful discussion and is a major caveat.

This is indeed a caveat. We added some text to discuss this (lines 569 to 576) drawing on other studies which have explored different methods for assigning protected area coverage. We also do our own exploration of alternative ways of defining protected area cover in an additional supplementary analysis, Supplementary Methods 2. Sanderson et al. 2021 shows that protected areas have benefits up to 5 km from the protected area, which supports our assumption that benefits will be visible at the 10 km square level. The fact that we find a positive effect of protected areas at the 10 km square scale also supports the scale we have used here. See also responses to comments 25 & 26.

14) Line 49. Surprising to see no reference to M Somveille's leading work on bird migration here in this introductory paragraph.

We focused mainly on references that referred to the Afro Palearctic migration system as that is our focus and we had to be ruthless in terms of which references we included to meet word limit requirements. Somveille's focus on energy fluxes is not directly relevant to our work on observed movements. Nonetheless, we now refer to a key paper that describes the drivers of migration in the introduction at line 60.

Somveille, M., Rodrigues, A. S., & Manica, A. (2018). Energy efficiency drives the global seasonal distribution of birds. Nature ecology & evolution, 2(6), 962-969.

15) Line 55. The comment: “many migratory species have been declining over recent decades, due to human activities such as habitat destruction, creation of barriers, over-exploitation and climate change” needs to be supported by more evidence and be more circumspect tone. Some of these are hypothesised and suggested, not proven causal or evidenced links.

We changed this sentence to emphasise that these effects aren't necessarily proven to: *This is thought to be due to human activities such as habitat destruction, the creation of barriers, over-exploitation and climate change^{6,7}, causing knock-on effects on ecosystem function and services⁵. We also added a reference to an additional review paper that looks at causes of declines in migrant birds (Vickery et al. 2023).*

16) Line 63. What is referred to as the “2010 United Nations Biodiversity Conference” was the 10th Conference of the Parties to the Convention on Biological Diversity held in Nagoya, Japan. This is where the ‘Aichi’ biodiversity targets come from. Perhaps clearer to use that language, if such targets are used at all.

We have changed this sentence as requested.

17) The Convention on Migratory species and CBD COPs are related, but quite separate processes. Does the CMS advocate for a percentage of the migratory species range that should be protected? That seems more relevant to your analysis than CBD PA targets.

Unfortunately, the CMS does not provide anything as specific as the percentage of a migratory species' range to be protected. Regarding protected areas, it focuses on advocating for:

- 1) co-ordinated conservation and management plans
- 2) conservation and, where required and feasible, restoration of the habitats of importance in maintaining a favourable conservation status, and protection of such habitats from disturbances, including strict control of the introduction of, or control of already introduced, exotic species detrimental to the migratory species
- 3) maintenance of a network of suitable habitats appropriately disposed in relation to the migration routes

We have now added more information into the introduction about approaches taken by other studies to set protected area targets for species and discussed the pros and cons of these, see our responses to comments 12, 18, 21, 23, 27, 28, 29, 34 & 36.

18) Line 67. When you say ‘need to be adequately protected’ what does that mean in a substantive sense? Has other work provided a guide or made suggestions? Theoretically, how much of a passerine or near passerine species' migrants' species range should ideally be covered by PAs? And on what basis? Presumably that might be different for other kinds

of migratory birds, such as congregatory waterbirds or seabirds? Do we know the answer to these questions? If not, how might it be reasonably estimated to inform your work?

No specific guidance is available on how much of either migratory or resident species' ranges should be covered by protected areas. We now describe previous approaches used to set species targets in more detail (see response to comments 12, 17, 21, 23, 27, 28, 29, 34 & 36), but we recognise that these approaches are not ideal. Rodrigues et al.'s 2004 method used arbitrary thresholds based on the global level of protected area cover at the time the paper was written which have no more biological basis than our targets of 17% or 30%. The method of protecting a larger area for smaller range species does have biological grounding and we have adopted this and redone our analysis to allow higher protection for periods of small range size in a similar way to Rodrigues et al. 2004. Most targets seem to be set at a country level, rather than a species level - presumably to enhance implementation feasibility. Mogg et al.'s 2019 method using IUCN criteria could be said to have more of a biological basis, but IUCN criteria were not made to be used in this way and using them in this manner caused so much controversy that the original paper suggesting this method was never published in a peer reviewed journal and exists as a pre-print only. The targets derived from this method are also unfeasibly large in many cases, which makes them impractical from a policy perspective. We felt that our analysis would be more useful to policy makers if we focused on existing, feasible targets. The 2020 and 2030 CBD targets were not made to refer specifically to species' ranges but, for widespread species (which all our study species are), if protected areas evenly represent 30% of the wider countryside to meet the 30 by 30 target, then we would expect ~30% of these widespread species' ranges to be protected, see lines 94 to 139 in the introduction, the methods at lines 579 to 586 and the discussion at lines 237 to 240 and 325 to 342.

We have shifted the focus of our paper away from meeting arbitrary targets and more towards looking at the existing % cover by protected areas for different species each week, as we feel this is probably more useful for conservation planners, see changes throughout the document.

19) Line 84. The ambition "By assessing protection on a weekly basis, we aim to account for migratory species' mobility to gain an accurate picture of the level of protection currently conferred and to uncover any gaps in the current protected area" is extremely laudable and would be new.

Thanks.

20) Line 98. There is live discussion as to whether the inclusive PA categories (V and VI) should be counted within 'PAs' as they may confer no conservation benefit as cultural living landscapes. Is there value in repeating your analysis without those categories?

To illustrate how much lower protection would be considered to be without these categories, we included another version of Figure 1a without category V and VI protected areas included, Figure 1b. We consider this further in the discussion (lines 357-363).

It is regrettable that >50% of PAs have no reported category, that makes understanding their impact even more difficult. It would be important to flag this issue in the discussion so that future research in this area might be improved.

We have flagged this issue now in the discussion (see lines 357-363). *Even when species are considered adequately represented by protected areas, protected areas vary substantially in the level of habitat management and protection present¹¹. For instance, the level of protection and management for protected areas in IUCN categories I to IV is substantially higher than for categories V and VI^{29,30}. The cover of protected areas for all species and weeks would decline substantially if we omitted protected areas in IUCN categories V and VI in our analyses (Figure 1b). This is despite the fact that IUCN designations are missing for over 50% of the protected areas. Considering the difference in protection conveyed by protected areas of different IUCN categories, infilling missing IUCN categories should be seen as a priority.*

21) Line 138. You state: “We found that, currently, no species was adequately covered by protected areas under the 2030 CBD target throughout the European stage of their lifecycle” but for reasons given above, that statement as is, is misleading.

We have changed the wording here to refer to our 30% target and explained the rationale behind these targets, and the limitations of target setting in general in much more detail, and focused more on the absolute percentage of protected area cover (see responses to comments 12,17,18,23,27,28 & 29).

22) Line 276. Agreed: “Here, we introduce an approach to assessing protected area coverage for mobile species. This approach could be refined to focus on particular types of protected area, or particular species, or expanded to assess a larger range of species and taxa where data are sufficient, though we have discussed certain caveats that need to be taken into account when making policy decisions.” There is genuine value and novelty in this analysis and data use, but I struggle with the current focus.

Thanks we hope our changes to the focus and additional explanatory text have helped with this.

23) Line 284. You state: “Our analysis found serious deficiencies in protected area cover for the 30 species of Afro-Palearctic migrant we assessed, we assessed, under both the 2020 and 2030 CBD targets” again, this is very questionable.

We have adjusted our language to make it clearer what we mean here: *Our analysis found a bias in protected area coverage where widespread species had a lower range coverage by protected areas than the wider landscape.*

24) Line 291. The quite novel methods are well described and carefully set out. However, I'd like to see some maps with the data coverage of the bird data and PAs to get a better sense of 'Europe' and better understand the analysis.

We have added maps with this information in the supplementary information (Supplementary Figure 4) and referred to in the methods at lines 492 and 562.

25) Line 387. The calculation of the proportion of a species' summed occurrence within a PA is the other part of this paper that needs some care, as stated by the authors, they do not know where in a 10km a species actually occurs and so whether that occurrence was in, or outside a PA. The approach taken is reasonable but this is somewhat inconsistent with the rather dogmatic and precise description of the results and their interpretation in the abstract

and discussion. It is touched on in the section on Limitations but not spelt out and emphasised. Were other approaches explored to deal with this issue or at least estimate sensitivity?

We now discuss in detail other ways we could have assigned square levels protection status see lines 567 to 574 in the methods. We also explore the impact of classifying a square as protected only if particular thresholds of (10% or 50%) a cell are protected. This is now included in the supplementary material (Supplementary Methods 2 and Supplementary Figure 3), referenced to at line 573 in the methods of the main manuscript and discussed at lines 420-440. See also our response to comment 13 and 26.

Reviewer #3 (Remarks to the Author):

This is an original manuscript examining the use of space and the use of protected areas by migratory birds (mostly passerines) across their annual life cycle while they are in Europe. Hence, including the breeding and the migration periods. The authors use citizen science data to model the distribution using a probability of occurrence and species distribution models including habitat/land use variables and topography. The SDMs were done at 10km² resolution across Europe.

26) The overall idea and analysis is original and new but the rationale is not fully clear or is not coherent. The scale of the study is not sufficiently detailed. It would be good to see more details on how small PAs (<10km²) were considered in this study. It may be that within an 10km² the bird distributions are concentrated in PAs (the effect of small PAs needs to be considered/discussed). It is possible that some species may have distributions concentrated in PAs but this is not detectable at 10km² resolution. I acknowledge that the nature of the data, collected as part of citizen science programmes may not enable a finer scale analysis (e.g. some of the complete species lists used may have been collected over large areas) which does not enable the authors to work at a smaller spatial scales. This is a problem that needs to be addressed from the perspective of the PA network. How many areas are less than 10km² and what was the approach adopted for small PAs?

It is possible that at a finer resolution (many protected areas are small and do not occupy a full 10km²) the species distribution may be more concentrated in PAs but the data and the analysis did not enable the authors to determine if this is the case.

Small protected areas were treated the same as large in this analysis. We followed the approach of *Araujo et al. 2011* to avoid the commission and omission errors that come from using an arbitrary threshold of protected area cover to categorise a grid square as protected or unprotected. We now discuss (lines 420-440) that our measure of species protection is based on an even distribution throughout a 10km square and could be improved upon in future if finer scale data become available or for future analysis for particular countries where finer-scale data may be available. We also added a supplementary analysis (Supplementary Methods 2), repeated with two different thresholds of 10% and 50% coverage to determine whether or not a 10 km square was protected, to assess the sensitivity of our results to these assumptions. This approach illustrates the influence on our quantitative findings of assumptions about the sub-10km relationship between protection and bird distributions. Despite this illustration, our approach best represents the relationship between protected area coverage and the current state of understanding of bird distributions, whilst avoiding the issues of arbitrary thresholds. See also response to comment 25 and 13).

27) 1- The authors investigate the use of protected areas (PAs) which, at present, correspond to 20.1% of Europe, by 30 migratory passerine and non-passerine species. The majority of the species are widespread (ranges >250,000km² and have 20% of their distribution in PAs (corresponding to the PA area available) as expected. what is adequate protection?

The answer to that is not clear cut. But we have explained our chosen targets in more detail now and the rationale behind them (see also responses to comments 12, 17, 18, 21, 23, 28, 29, 34, 36)

28) 2- PAs have a variety of criteria behind their classification, while some were classified to protect endangered bird species others may have been designated due to landscape features or habitats and may not be particularly effective at protecting migratory species. In most countries PAs have natural or semi-natural landscapes and will be effective at protecting the species that depend on those features, not the wider countryside species. The study explores if the current PA network adequately protects migratory species but the network was not built for this purpose. this should be better explained.

We have now explained our rationale for these targets better and some of the limitations of target based conservation approaches, please see lines 94-139 in the introduction 579-586 in the methods and lines 237 to 240 and 325 to 342 in the discussion and responses to comments 12, 17, 18, 21, 23, 36 & particularly 29 and 34). We have moved the focus of the paper more towards the absolute percentage of protected area cover.

29) 3- The Natura 2000 SPAs were classified to protect endangered bird species and many migratory species included in this study were not part of the priority list. The rationale for the study is not fully justified, why should the PA network covering 20% of the land protect more than 20% of the distribution of the migratory species, if this was the case it would indicate that migratory species were becoming concentrated in PAs disappearing from the wider landscape, which is not desirable. Overall the rationale of the paper is poorly explained, it is preferable to find that migratory species are still found across large areas rather than concentrated on PAs (having said this the scale of the study makes it difficult to conclude this with certainty).

The rationale for the study needs to be better explained.

Sorry that this was unclear. We intended that adequate protection was when these widespread migratory species were covered by protected areas to the same extent that the 2020 and 2030 CBD targets required the wider landscape to be. So we meant that under the 2020 CBD target, 17% of the species' range should be protected each week of the year, as the wider landscape should be. Under the 2030 CBD target, this changes to 30% for both the widespread species' range and the CBD target. We have now clarified this in the text (see our response to comments 12,17, 18, 21, 23, 27 and 28).

30) 4- The modelling procedure, done on a weekly basis, and including the migration period may include birds migrating and not using habitat, hence modelling habitat use during migration may not be fully appropriate. This should be justified.

The EBP data will likely only include birds actively using the habitat. Our study species, which are small Afro-Palaearctic migrants, generally migrate at night (see Dorka, V. Das jahres- und tageszeitliche Zugmuster von Kurz- und Langstreckenziehern nach

Beobachtungen auf den Alpenpässen Cou/Bretolet (Wallis). Der Ornithol. Beobachter 63, 165–223 (1966)) and it is also unlikely that our focal small passerines will be recorded when just passing over on migration. Some studies have found particular habitat associations of birds during active migration (Zuckerberg et al. 2016, La Sorte et al 2022). We added the following text to the methods lines 502-504:

The EBP data will likely only include birds actively using the habitat as our study species generally migrate at night⁶⁵, and other studies have found habitat associations with migrants in active migration^{66,67}.

31) 5- It is interesting to note that during migration many species have very low representation in PAs which suggests that there are very few PAs in migratory corridors which could have been better discussed by the authors.

We don't agree with this statement for spring migration, as spring migration actually had the highest overlap with protected areas overall. Autumn migration on the other hand does appear to be less well presented by protected areas, though not significantly less than the breeding season. We have added the following text to the discussion, lines 293-297, although we are not considering the tail ends of migration here, as we only include weeks when 25% or more of the maximum summed occurrence of a species is in Europe.

Overlap with protected areas seems lower during autumn passage than at other times of year (though not significantly lower than in the breeding season), possibly suggesting a lack of protected area cover in autumn migratory corridors, when mortality may be greatest⁴⁴. Of course, a species may only be transiently in autumn passage areas, and therefore, temporary measures – such as seasonal hunting bans⁴⁵ or grazing restrictions – could be effective for passage areas.

32) 6- The authors find that species that have higher representation of their distributions in PAs are doing better (likely species benefitting from conservation actions and habitat management measures) than those that occur in the wider countryside. This is not new.

We believe this finding is novel, although this was not a focus of our analysis, it is novel in the context of the migratory species considered here and the 10 km resolution and continental scale of our analysis. The fact that we found a positive effect of protected area cover is useful in that it corroborates findings from other work on a broader set of species (e.g. Donald et al. 2007, Barnes et al. 2021), but is particularly informative given the long-term decline in migratory species for which there is ongoing debate about the most effective solutions to their conservation (Vickery et al. 2023). We have added the following text to the discussion at lines 331 to 334::

However, the positive association between protected area coverage and long-term population trend provides important new evidence that extending protected area coverage in Europe is likely to benefit declining long-distance migratory birds, assuming the past is a good predictor of the future.

33) 7- Eurasian Hoopoe and Ortolan Bunting seems to have less than expected distribution in PAs, across the whole time they spend in Europe, this suggests that the PA network is particularly poor for these species. This is an interesting result that could have been discussed in more detail.

We have now added some text to discuss this (lines 264-276):

*Of the species we assessed, the Ortolan Bunting *Emberiza hortulana* was the least well covered by the protected area network by a considerable margin. Its western flyway population is threatened by*

*hunting and agricultural intensification on the breeding grounds, and flyway-scale conservation measures are required³⁹. Eurasian Hoopoe *Upupa epops* was the next least well covered species (Figure 1) and both species are primarily agricultural specialists^{40,41}, a habitat that the protected area network does not represent well^{33,35,37}. Alternative conservation approaches, such as a land sharing approach integrated into farming^{42,43}, may hold more promise for these species.*

34) 8 - To achieve the 2030 CBD target we would need to classify 30% of the planet, and in this case we would expect to protect 30% of the distribution of these migratory species. It is not expected or desirable that the current 20% of Europe that is classified as PAs should have 30% of the distribution of these widespread species (it would mean these species would no longer be widespread across Europe).

Apologies we were not clear. We were not suggesting that the current 20% cover of protected areas should target 30% of a species' range. We were looking at how far there was to go in the wider landscape to reach the 2030 target, considering we are almost half way through the current decade. As these species are widespread, they can help to indicate biases in the protected area network, when the percentage of their range protected differs from that of the wider countryside. We have now moved the focus of the paper away from the 2020 and 2030 targets and more towards the percentage of protected area cover per week and we have redone Figure 1 to reflect this. Please also see our above responses to similar comments (12, 17, 18, 21, 23, 28, 29, 36).

Methodological issues

35) 8- The choice of the 30 species included in the study is not justified.

We have now included a line in the methods to explain this in the source data section lines 468-470:

These 30 species were chosen as they were all the passerine and near passerine species included in EuroBirdPortal at the time.

36) 9- What do you mean by adequately protected (e.g. line 158)? Would the authors expect that more than 17% (CBD2020 target) of the distribution should be included in PAs? This was not fully explained.

We have now changed the wording here to make it clearer what we mean by adequate protection and that the targets we consider are arbitrary. We have explained some of the difficulties in determining what is meant by 'adequately protected'. See above responses to similar comments (12, 17, 18, 21, 23, 28, 29, 34).

RESPONSE TO REVIEWERS' COMMENTS

Reviewer #1 (Remarks to the Author):

We acknowledge the effort done by the authors of this contribution and are happy with its current status

Reviewer #2 (Remarks to the Author):

Very pleased to see this revised manuscript and the care taken in responding to my original comments and concerns. The revised manuscript is much improved and reads very well. I have no substantive comments but some suggestions and observations below.

Incidentally, it did occur to me that you could use this analytical framework to specifically identify and map priority areas in Europe for area protection for migratory landbirds towards 30x30, in a more propositional way, perhaps that is your next paper.

Thanks, yes that's a good idea.

Line 105 & 179. The species here are essentially terrestrial species, landbirds, so it would be good to make that clear upfront. You aren't including seabirds, waterbirds, raptors etc that readers might think of immediately as typical migratory, congregatory birds.

We have added a few words to make it clearer we mean passerine and near passerine landbirds (lines 98-99 and 177-178):

We modelled the weekly distributions of 30 species of passerine and near passerine Afro-Palaearctic migrant landbirds during their occurrence in Europe.

Line 202. Here and elsewhere, meeting an area target for PAs tells us nothing about site quality and management as you touch on later and isn't strictly the CBD target. The CBD targets have area, plus 'effectively conserved and managed....etc etc'. It would be important to stress the two parts to the CBD targets – more 'paper parks' are likely to be of little use to conservation.

We completely agree about needing effective management. Which is why we discuss it in a later paragraph. However, in this sentence we just talk about the area target being achieved. It doesn't really work with the flow of the manuscript to add a bit about management to that line in question. However, we have added in a bit about this later on in the discussion to make it clear that we're just discussing the area target. We don't have to data to determine whether the management met the target (lines 307-309):

The 2020 and 2030 CBD targets also stipulated that the 17% or 30% protected area cover should be conserved through effective management, but that was not something we had the data to assess in this study.

Line 249 – Could the lower overlap with protected areas during autumn passage be a product of observer recording behaviour. All birders are keen to see and record the first spring X species, but I wonder by autumn whether recording enthusiasm is as good and departing birds are as well recorded? Any comment or data to check on this?

Lower observer effort in autumn may affect the casual lists but would not affect our complete list data. Though even if people were less likely to record migrants in autumn, this would not affect the percentage cover of protected areas, it would just mean the whole distribution had a lower probability of occurrence. We see no reason why recording effort would be biased so that migrants in protected areas were less likely to be recorded. Also, autumn is still a popular time for birdwatching due to the possibility of rare migrants blown off course and there was not actually a significant difference between protected area cover in autumn passage and in the breeding season.

Line 178 – While the Discussion is good it is very long. It seems to me that some of the very detailed ‘caveat’ paragraphs here might be better placed in the supporting materials to allow a better flow for the reader.

While we agree that the discussion is long, it does fit within the word limits of Nature Comms and we are reluctant to move some important caveats researchers should be aware of to Supplementary Information and risk the researcher not reading these and therefore making conclusions without all the facts.

Richard Gregory

Reviewer #3 (Remarks to the Author):

This manuscript makes use of a strong temporally and spatially explicit dataset. The authors use information from the EuroBird Portal containing distribution information for 30 migratory European African passerine and near-passerine species, on a weekly basis, to assess the time birds spend in protected areas across their life cycle. This is a robust 10-year dataset with weekly information for the 30 migrant species. The authors modelled the weekly distribution using habitat, climate and elevation information obtained at the 10km grid reference using spatiotemporal exploratory models.

The authors addressed most of my previous comments and I am happy with the new version of the manuscript.

A few last comments/queries:

The authors mention “For protected areas to be effective in conserving migratory species, key habitats and resources at all stages of their life cycles need to be adequately protected” this is true but the current Natura 2000 network does not target many of the species included in this manuscript so we would not expect these species to be well represented. Revising the species in Annex I of the Birds Directive and including more migratory and farmland species should be made a priority?

We had added a sentence into the discussion to make this point, lines 262 to 264: *Revising the Annex 1 of the Birds Directive to include more migratory and farmland species would help to make these species a higher priority when designating protected areas.*

The authors also state in line 331: “However, the positive association between protected area coverage and long-term population trend provides important new evidence that extending protected area coverage in Europe is likely to also benefit declining long-distance migratory birds, assuming the past is a good predictor of the future”. As mentioned earlier in the

manuscript, many of the species that are declining are farmland birds (not targeted in Annex1 of the Birds directive so not prioritised for conservation in PAs). The declining species are associated with agricultural landscapes that suffered transformations due to the intensification of agriculture. Classifying agricultural land as protected areas may help these species due to increased adoption/use of agricultural environmental policies. Would more funds be available to farmers in these areas to prevent further intensification of farmland? Taking into account funds for conservation are limited, setting up priority areas for farmland/migratory birds would concentrate resources in targeted areas instead of promoting horizontal measures (e.g. wildflower edges or parcels) across wider landscapes. I realise you are not yet sure which approach should be used for which species (line 340) but the consequence of more protected areas, and protected areas for the species in this study suggests you are proposing the use of more resources in targeted areas rather than more horizontal measures across the landscape. If this is not your recommendation you should make it clear.

We are saying that current protected area coverage for migratory birds does not match recent international commitments, and that given the positive association between protected area coverage and long term population trends, addressing that coverage gap is likely to have benefits for migratory bird conservation. That does not mean that this is necessarily the best use of conservation resources, for farmland birds, prioritising wider agri-environment interventions may be more appropriate in some circumstances, even outside of protected areas. However, as the reviewer argues, expanding protected areas for migrants into farmland habitats may provide a good way of targeting that intervention most appropriately for those species, although that is not a specific intervention modelled in the paper. We've added some text to make this point more clearly. See lines 230-233:

Eurasian Hoopoe *Upupa epops* was the next least well covered species (Figure 1) and both species are primarily agricultural specialists^{40,41}, a habitat that the protected area network does not represent well^{33,35,37}. Alternative conservation approaches, such as a land sharing approach integrated into farming^{42,43}, or increasing protection of farmland as a mechanism for promoting bird-friendly farmland management in these areas, may hold more promise for these species.

We haven't looked at the benefits of agricultural environmental schemes here as it's outside the scope of our analysis and therefore we cannot advise specifically on whether it is more beneficial than protected areas for farmland species.

Minor comments:

Line 1003 lacks a T in The colours Done

Supplementary Methods 4 Environmental Variables:

What do you mean with the sentence: "For the model to model complete list length against on space, time and habitat, we used Copernicus Global Land Service 2015 Land Cover data at a 100 m by 100 m resolution"

We have re-phrased this to be clearer: *For the model to determine pseudo-complete lists from casual records (see below), we used Copernicus Global Land Service 2015 Land Cover data at a 100 m by 100 m resolution (<https://land.copernicus.eu/en/products/global-dynamic-land-cover>).*

Reviewer #4 (Remarks to the Author):
